# Can Small Language Models be Good Reasoners for Sequential Recommendation?

## ABSTRACT

Large language models (LLMs) open up new horizons for sequential recommendations, owing to their remarkable language comprehension and generation capabilities. However, there are still numerous challenges that should be addressed to successfully implement sequential recommendations empowered by LLMs. Firstly, user behavior patterns are often complex, and relying solely on one-step reasoning from LLMs may lead to incorrect or task-irrelevant responses. Secondly, the prohibitively resource requirements of LLM (e.g., ChatGPT-175B) are overwhelmingly high and impractical for real sequential recommender systems. In this paper, we propose a novel Step-by-step knowLedge dIstillation fraMework for recommendation (SLIM), paving a promising path for sequential recommenders to enjoy the exceptional reasoning capabilities of LLMs in a "slim" (*i.e.,* resource-efficient) manner. We introduce CoT prompting based on user behavior sequences for the larger teacher model. The rationales generated by the teacher model are then utilized as labels to distill the downstream smaller student model (e.g., LLaMA2-7B). In this way, the student model acquires the step-by-step reasoning capabilities in recommendation tasks. We encode the generated rationales from the student model into a dense vector, which empowers recommendation in both ID-based and ID-agnostic scenarios. Extensive experiments demonstrate the effectiveness of SLIM over state-of-the-art baselines, and further analysis showcasing its ability to generate meaningful recommendation reasoning at affordable costs.

## KEYWORDS

Recommender Systems, Large Language Models, Distillation

**ACM Reference Format:**
Anonymous Author(s). 2018. Can Small Language Models be Good Reasoners for Sequential Recommendation?. In *Proceedings of Make sure to enter the correct conference title from your rights confirmation emai (Conference acronym 'XX).* ACM, New York, NY, USA, 10 pages. https://doi.org/XXXXXXX.XXXXXXX

**Statement of Relevance.** This paper focuses on user behavior in the context of the web and designs a large language model based recommender system, which is highly relevant to the "User Modeling and Recommendation Track", an essential **Web Application**.

## 1 INTRODUCTION

Sequential recommendation is extensively utilized in a variety of internet applications due to its prominent performance in uncovering a user's evolving and dynamic interests from his/her chronological interactions [20]. Despite the effectiveness, existing models are often trained on a closed-loop user-item interaction dataset, inevitably suffering from severe exposure bias and popularity bias. Therefore, beyond narrow information present in the original datasets, it is crucial to incorporate open-world knowledge to foster a more comprehensive and generalized understanding of historical behaviors.

Due to the impressive reasoning capability, the recent emergence of Large Language Models (LLMs), such as GPT 3.5/4, has brought a significant breakthrough in various NLP tasks [18, 19, 28, 35], showing substantial potential in overcoming the isolated nature of real-world sequential recommenders that rely on closed data sources for training [1, 22]. These LLMs are trained on massive corpora, granting them to exhibit the remarkable capability of human-like thinking as well as seamless reasoning. Roughly speaking, current LLM empowered recommenders mainly fall into the following two groups: (1) **LLM as a ranker**, which typically involves prompting the frozen LLM to offer a reasonable ranked list that satisfies the user interests [9]. However, solely relying on the zero-shot or few-shot learning capability of LLMs is still inferior compared to traditional sequential recommendations that utilize in-domain collaborative knowledge. To address this limitation, (2) **LLM as a knowledge enhancer** has been proposed, typically following a cascading architecture: the LLM is first instructed to generate rich knowledge (*e.g.,* user preference and factual knowledge on items), followed by a classical recommendation backbone for harvesting in-domain knowledge and collaborative signals. Generally, the bridging of both worlds tends to elicits a more promising performance [31]. While LLMs for recommendation hold promise, they also face significant challenges that cannot be ignored.

*One is the exceptional reasoning capability of LLM within the context of recommendation has not been fully explored.* There is a gap between the open-world nature and recommender systems, which means that the recommendation knowledge generated by LLMs may be incorrect or task-irrelevant. Fortunately, with the chain-of-thought (CoT) prompting strategy [10, 17, 27], LLMs can break down complex tasks into a series of intermediate reasoning steps, which can improve the ability to understand behavior patterns and explore user interests. Consequently, there is a strong motivation to leverage the CoT reasoning capability of LLMs in sequential recommender systems, enabling the generation of targeted recommendation-related rationales. For instance, guiding LLMs to reason progressively, similar to a human salesperson, to deduce user interests, narrow down the categories of items that align with their interests, and ultimately recommend specific items within these categories that the user is likely to interact with.

*Another significant challenge is the prohibitively high resources are far beyond affordable for real-world recommender systems.* The immense size of LLMs demands a considerable amount of memory and computational power, which necessitates specialized infrastructure. For instance, the deployment of the open-source LLaMA2-70B requires eight Nvidia A100 servers. On the other hand, working with closed-source LLMs also involves significant costs. For instance, using ChatGPT as an example, the current approach requires calling its API, which comes with substantial monetary expenses. For instance, in gpt-3.5-turbo, the costs are approximately $0.0015 per 1,000 tokens for input and $0.002 per 1,000 tokens for output. Therefore, a natural question arises:

*Can a language model with affordable costs still serve as an effective reasoning engine for sequential recommendation?*

To answer this question, in this paper, we propose a novel S̲tep-by-step know̲L̲edge d̲I̲stillation fra̲M̲ework for recommendation (**SLIM**), which enables sequential recommendations to enjoy the significant reasoning capabilities of LLMs in a "slim" (*i.e.,* resource-efficient) manner. Specifically, we develop a step-by-step knowledge distillation strategy for sequential recommendations to transfer the reasoning capabilities of LLMs (*i.e.,* teacher) to a small language model (*i.e.,* student). This strategy guides the larger teacher model to engage in macro-to-micro thinking for complex recommendation task through CoT prompting. Through the process of distillation, the small student model with only 4% parameters of the large teacher model acquires step-by-step thinking capabilities and evolves into a good reasoner. Subsequently, we directly deploy the small language model as a knowledge generator for sequential recommendation, which can derive high-quality reasoning knowledge highly relevant to recommendation. These knowledge reflect user preferences for categories, brands, and specific items, which can be flexibly integrated with any sequential recommendation backbone, including ID-based and ID-agnostic scenarios. Our key contributions can be summarized as follows:

- To the best of our knowledge, it is the first knowledge distillation framework of LLMs tailored for sequential recommendation.
- We propose SLIM, a novel step-by-step knowledge distillation framework, empowering sequential recommenders with the CoT reasoning capabilities of LLMs in a resource-efficient manner.
- Extensive experiments on three datasets demonstrates the effectiveness of our proposed SLIM. Further analysis reveals that SLIM generates meaningful reasoning at affordable costs.

## 2 THE PROPOSED FRAMEWORK

In this section, we propose **SLIM**, a novel knowledge distillation framework tailored for recommendation, which incorporates the reasoning capabilities of LLMs into recommender systems in a resource-efficient manner. The overview is illustrated in Figure 1.

### 2.1 Sequential Recommendation Backbone

Sequential recommendation aims at the accurate prediction of users' next behavior by capturing evolved and dynamic preferences over historical behavior sequences, which has occupied a critical position in various modern information systems [20]. In general, the success of sequential recommendation typically hinges on the meaningful representation of items and effectively encoding behavior patterns.

**Item Representation.** For neural sequential recommendations, the item encoder is the key component which transfers the items to representations. Formally, given an item set $\mathcal{I}$, each item $i$ may be associated with several optional attributes $f_i$, such as title, category and brand. The encoder can generate the item representations for each item based on their ID (*i.e., i*) and attributes (*i.e., $f_i$*):

$$z_i = \text{ItemEncoder}\left(i, f_i\right), \tag{1}$$

where $z^i$ is the representation of item $i$. Generally, ItemEncoder($\cdot$) is implemented as a hybrid architecture where a embedding layer aims at tackling ID-like features (*e.g.,* item id), coupled with a text encoder (*e.g.,* BERT [4]) for context embedding based on the item description (*e.g.,* title, category).

**Sequential Encoding.** To capture the sequential characteristics of user behaviors, the action sequence of user $u \in \mathcal{U}$ can be organized in chronological order $\mathcal{S}_u = [i_1, i_2, \cdots, i_{t-1}]$, where $i_k \in \mathcal{I}$ represents the $k$-th item that the user $u$ interacted with. Next, each item in $\mathcal{S}_u$ is firstly fed into ItemEncoder($\cdot$) (denoted as $\mathcal{Z}_u$), followed by the a sequential encoding.

$$s_u = \text{SeqEncoder}\left(\mathcal{Z}_u\right), \tag{2}$$

where $s_u$ denotes the representation of sequence $\mathcal{S}_u$. SeqEncoder is sequence encoder, which can be implemented with the Attention [25] or other neural architectures [21, 34]. Based on the sequence $\mathcal{S}_u$, our objective is to predict the next item $i_t$ that the user $u$ is likely to interact with at the $t$-th step.

**Prediction and Optimization.** After generating the above representations, we can obtain the final prediction $\hat{y} \in \{0, 1\}$ at time $t$ with dot product or MLP layer followed by a sigmoid activation function [6], where each element $\hat{y}_{ui}$ indicates how likely the item $i$ should be recommended to the target user $u$. Finally, the model is trained with binary cross-entropy [3] loss as follows:

$$\mathcal{L} = -\sum_{u \in \mathcal{U}} \sum_{i \in \mathcal{I}} y_{ui} \log \hat{y}_{ui} + (1 - y_{ui}) \log \left(1 - \hat{y}_{ui}\right). \tag{3}$$

Note that these classical sequencical models typically perform recommendation based on the user action sequences and item attributes (*e.g.,* title, category and brand), lacking the reasoning power that have recently emerged in LLMs.

## 2.2 Step-by-Step Knowledge Distillation for Recommendation

Despite the remarkable reasoning ability of LLMs, it is non-trivial to adapt LLMs to empower the traditional recommender systems. The challenge arises from two aspects: (1) Complex behavior patterns of users are difficult to understand directly by LLMs. (2) The large size and high inference latency of LLMs exacerbates resource-consuming. Therefore, we propose step-by-step knowledge distillation to transfer the reasoning capabilities of LLMs to a smaller LLaMA2-7B [24] model specialized for the recommendation tasks.

In detail, our distillation strategy consists of two straightforward steps: Firstly, we employ CoT prompting related to user behavior to guide the LLM (*i.e., teacher*) in thinking step-by-step and generating natural language rationales that support its predictions in the recommendation scenario. Secondly, these rationales are subsequently utilized as labels to fine-tune the downstream smaller language

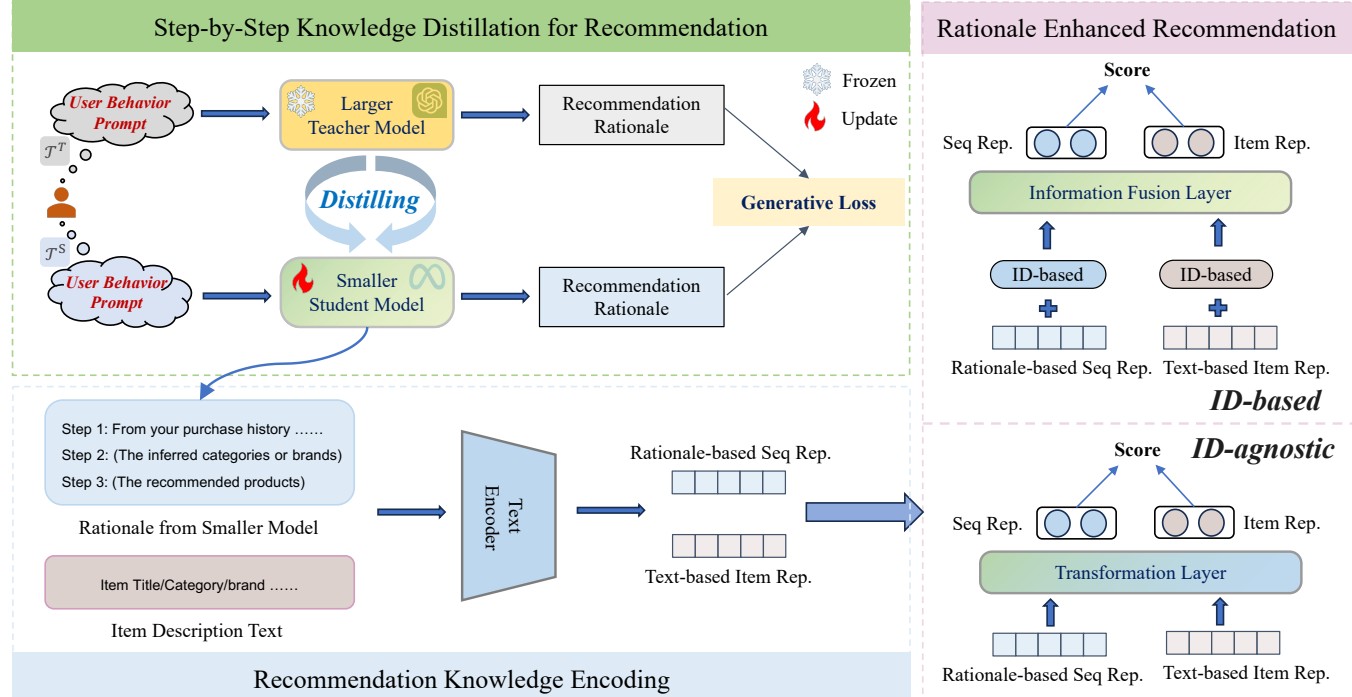

Figure 1: The overview of the proposed framework.

model (*i.e., student*), enabling it to approach the reasoning capabilities of the larger model in the recommendation domain. Finally, the fine-tuned smaller model acts as the ultimate knowledge generator, offering reasoning knowledge to the recommender systems.

*2.2.1 Extracting Recommendation Rationales from LLMs.* Traditional sequential recommendations rely solely on scores to perform recommendation and ignore the intermediate reasoning steps, which constrains the accuracy and explainability of recommendations. Inspiringly, the emergence of LLMs has led to a significant breakthrough in understanding human and generating rationales step-by-step for recommendation through CoT prompting. Therefore, we guide LLMs in generating critical reasoning, encompassing user preferences, interested categories/brands and specific items, all of which are essential for providing appropriate recommendations.

To achieve this objective, we utilize zero-shot CoT prompting to elicit LLMs in extracting this reasoning information from user behaviors. Specifically, given the user set $\mathcal{U}$ and the behavior dataset $\mathcal{D} = \{\mathcal{S}_u | u \in \mathcal{U}\}$, we construct a user sub-set $\mathcal{U}'$ and a small behavior sub-dataset $\mathcal{D}' = \{\mathcal{S}_i\}$ by random sampling, where $\mathcal{U}' \subset \mathcal{U}$ and $|\mathcal{U}'| \ll |\mathcal{U}|$. Then, we have meticulously designed a CoT prompt template $\mathcal{T}_t$ to facilitate in-depth reasoning according to user behavior by LLMs. As illustrated in Figure 2, our instructions consists of three progressive steps:

- *Step*1. Summarize user preferences based on the historical behavior sequences.
- *Step*2. Recommend categories or brands to the user based on the summarized preferences.

- *Step*3. Recommend products that align with the recommended categories/brands.

The template starts with a macro perspective and gradually zooms in to a micro perspective, which guides LLMs in a step-by-step thinking process and ensures that the output of the LLM conforms to a specific format. Through this process, LLMs can effectively leverage extensive open-world knowledge to infer the aspects that users are likely to be interested in the future. As shown in Figure 2, through CoT prompting, the teacher model can generate informative recommendation rationales in response.

Subsequently, we further fill the template $\mathcal{T}_t$ with user historical behaviors in small dataset $\mathcal{D}'$ to generate corresponding CoT prompts $\mathcal{X}' = \{x'_u | u \in \mathcal{U}'\}$ for the teacher LLM model. With this incremental CoT prompts $\mathcal{X}'$, LLMs will generate corresponding recommended rationales $r'_u \in \mathcal{R}'$ for each input $x'_u$. Although the current recommendations overlook these rationales, they are crucial for achieving more efficient and explainable recommender systems. Besides, unlike previous studies [31], we do not concentrate on inferring item factual information (*e.g.,* the director of movie items), as most real-world items (*e.g.,* food, home and kitchen items) are difficult to associate with trustworthy open-world knowledge.

*2.2.2 Fune-tuning Smaller Models with Recommendation Rationales.* By guiding the thought process of LLMs step-by-step, we can comprehend complex behavior patterns of users and generate high-quality recommendation rationales. However, their large scale and computational overhead make them unsuitable for recommendation scenarios that require low latency. For instance, serving a single 175 billion LLM necessitates a minimum of 350GB of GPU memory [36].

---

**Template ($\mathcal{T}_t$) for Teacher Model**

I've purchased the following products in the past in order: **<Historical Behaviors>**
Please help me to do the following things in order:
Step 1: Could you help me identify the key factors that influence my choice of products by analyzing my purchase history (summarize my preferences briefly)? Let's work this out in a step by step way to be sure we have the right answer.
Step 2. You will select the product categories or brands that appeal to me the most based on my personal preferences. Please split these output with a line break (Format: no. a product category or brand).
Step 3. Based on my purchase history, can you recommend 5 products that fit the categories or brands selected in Step 2? Please split these recommended products with a line break (Format: no. a recommended product).

---

**Historical Behaviors**

0). Xbox 360 Live 12 Month Messenger Gold Pack, 1). Xbox 360 Live Camera Bundle, 2). The Sims 3 Deluxe [Download].

---

**Teacher Model Response**

Step 1: Based on your purchase history, it seems like you are interested in gaming-related products that focus on online gaming and social interaction. You also seem to be interested in simulation games.
Step 2:
1. Xbox Live products
2. Sims franchise
Step 3:
1. Xbox Live Gold Membership
2. Xbox Wireless Controller
3. The Sims 4 Deluxe Edition
4. The Sims 4 Cats and Dogs Expansion Pack
5. The Sims 4 City Living Expansion Pack

---

**Figure 2: Zero-shot CoT prompting for larger teacher model. Eliciting LLMs to generate recommendation rationales in a step-by-step manner.**

Despite recent study [31] attempting to mitigate this issue by off-line inference, it's still unaffordable to generate recommendation rationales for all users in the real-world scenario.

To this end, we leverage knowledge distillation to transfer the recommendation reasoning capabilities of larger teacher models to smaller student models, thereby reducing the computational overhead. Considering that complex prompts can improve the reasoning quality of large models but greatly increase the understanding difficulty of small models, we design simplified template $\mathcal{T}_s$ based on the template $\mathcal{T}_t$, as showed in Figure 3. Subsequently, we generate simplified prompts $\mathcal{P}' = \{p'_u | u \in \mathcal{U}'\}$ as input, and collect the rationales $\mathcal{R}'$ generated by teacher LLMs as the expected output labels to fine-tune the smaller student model. As a result, for a given input instruction $p'_u$, we train the smaller model with parameters $\theta$ to generate the corresponding recommendation rationale $r'_u$. Formally, we optimize the negative log-likelihood of conditional language modeling objective as follows:

$$\mathcal{L}_{distill} = \sum_{u \in \mathcal{U}'} \sum_{t=1}^{|r'_u|} \log \left( P_\theta \left( r'_{u,t} \mid p'_u, r'_{u,<t} \right) \right), \quad (4)$$

where $r'_{u,t}$ is the $t$-th token of the $r'_u$, $r'_{u,<t}$ represents the tokens before $r'_{u,t}$. To conserve resources, we employ the LoRA [11] for parameter-efficient model fine-tuning. This approach involves training only a small set of additional parameters instead of the entire model. Through experimental validation, we demonstrate that the generated rationales maintain a comparable quality to models with 25 times the model size, despite using a limited number of training samples and a smaller model size. As illustrated in Figure 3, the student model responses showed a step-by-step reasoning ability similar to that of the teacher model. For instance, the student model initiates by logically inferring the user's intent by leveraging its recommendation-related CoT. Subsequently, it offers potential

---

**Template ($\mathcal{T}_s$) for Student Model**

I've watched the following movies in the past in order: **<Historical Behaviors>**
Please provide a brief summary of my products preferences, followed by the recommendation of five product categories or brands that align with my interests, and then suggest five products that fall under each category or brand.

---

**Historical Behaviors**

0). Gundam Breaker, 1). Shin Gundam Musou, 2). Forza Motorsport 4 - Xbox 360, 3). Armored Core: For Answer - Xbox 360, 4). Dynasty Warriors: Gundam 3 - Xbox 360, 5). Import Tuner Challenge - Xbox 360, 6). Forza Horizon - Xbox 360, 7). PlayStation Vita Memory Card 64GB (PCH-Z641J).

---

**Student Model Response**

Step 1: From your purchase history, it seems that you enjoy gaming-related products that are action-packed, have fast-paced gameplay, and involve warfare or fighting. You also seem to prefer games based on popular franchises such as Dynasty Warriors and Gundam. Additionally, you have a preference for racing and sports games.
Step 2:
1. Racing games
2. Action-packed games with fast-paced gameplay
3. Games based on popular franchises such as Dynasty Warriors and Gundam
Step 3:
1. Forza Horizon 5 - Xbox Series X/S
2. Gundam Breaker: Battlege - PlayStation Vita
3. Dynasty Warriors 9 - PlayStation 4
4. Need for Speed Heat - Xbox One
5. Gran Turismo Sport - PlayStation 4

---

**Figure 3: Prompting for fine-tuned student model. Eliciting smaller model to generate recommendation rationales in a step-by-step manner.**

game genres that align with the user's interests. Ultimately, several specific games are recommended to the user.

Overall, by utilizing recommendation rationales as labels instead of generating pseudo-labels for recommended results from LLMs, we enhance the smaller language model with step-by-step reasoning capabilities similar to the reasoning process of the larger model.

## 2.3 Empowering Recommender with Reasoning Knowledge

With the help of step-by-step knowledge distillation, small language models can become efficient reasoners. However, traditional sequential recommendation models cannot directly utilize the rationales of natural language forms. Thus, in this section, we explore how to apply the recommendation rationales generated by small language models to the sequence recommendation model, enabling it to efficiently combine the reasoning ability of LLMs in a resource-efficient manner. Specifically, we introduce two application approaches. The first approach is ID-based, where we treat the rationales text as supplementary knowledge and combine them with ID-based recommendation backbone to improve the traditional closed-loop learning dependent on user-item interactions. The second approach is ID-agnostic, where we encode the rationale text of user behaviors and the description text of candidate items as the representations of user and item, respectively. This allows us to make recommendations based on text similarity.

*2.3.1 Encoding Recommendation Rationales.* Owing to the efficient reasoning power of small language model, each user behavior sequence $\mathcal{S}_u \in \mathcal{D}$ can be associated with corresponding CoT rationales $r_u$, while each item $i$ can be associated with attribute descriptions $f_i$ (*e.g.,* title, category, brand). Then we leverage pre-trained language models (PLMs) to learn text representations, enabling the measurement of semantic distance in vector space. Concretely, we adopt the text encoder to map the text on both the item side and

the sequence side into a unified semantic space:

$$z_i^{\text{text}} = \text{TextEncoder}(f_i),$$
$$s_u^{\text{text}} = \text{TextEncoder}(r_u), \quad (5)$$

where $z_i^{\text{text}}$ and $s_u^{\text{text}}$ represent the text representations of item descriptions and recommendation rationales, respectively. The encoder TextEncoder$(\cdot)$ can be flexibly configured as any frozen or trainable text encoding model, which we instantiate with BERT [4] in this work. Due to the step-by-step thinking process of language model, the representation $s_u^{\text{text}}$ encodes rich reasoning knowledge from open-world at both macro-level (*i.e.,* general user preference) and micro-level (*i.e.,* specific recommended item).

*2.3.2 Utilizing Recommendation Rationales.* Traditional sequential recommendation methods learn the user sequence representation based only on the user-item interaction history, resulting in an information-enclosed model. To alleviate this issue, we enhance the traditional recommender systems with the rationale representations obtained from Eq. (5), which is derived from open-world knowledge and deep reasoning about the user's behavior patterns. Specifically, we leverage it through the following two approaches.

**Empowering ID-Based Recommendation.** To disrupt the closed systems of sequential recommendation, we integrate the rationale representations into traditional recommendation backbone, effectively combining the open-world reasoning knowledge with the collaborative signal of traditional recommendations. Specifically, we propose an information fusion layer to combine the meaningful text representations (*i.e.,* $s_u^{\text{text}}$ and $z_i^{\text{text}}$) with the original embeddings in the backbone model as follows:

$$z_i = g_f([g_l(z_i^{\text{text}}); z_i^{\text{id}}]),$$
$$s_u = g_f([g_l(s_u^{\text{text}}); s_u^{\text{SeqEnc}}]), \quad (6)$$

where $z_i^{\text{id}}$ is the ID embedding of item, $s_u^{\text{SeqEnc}}$ is the sequence representation obtained from SeqEncoder (*i.e.,* Eq (2)) in backbone model, $[;]$ denotes the concatenation operation, $g_l(\cdot)$ transforms the text representations to the same dimension with the ID embeddings, and $g_f(\cdot)$ is an fusion layer that enables the model to learn and incorporate flowing information from both sources. Without loss of generality, we implement $g_l(\cdot)$ and $g_f(\cdot)$ with linear layers.

**Empowering ID-Agnostic Recommendation.** Recent studies have revealed that sequential models that focus on text modeling exhibit superior generalization abilities and are more effective in handling cold-start items [8, 13]. Therefore, we explore a direct utilization of rationale representations in ID-agnostic recommendation scenarios. In this case, the representations of item text and rationale are directly transformed into a unified space as follows:

$$z_i = g_t(z_i^{\text{text}}),$$
$$s_u = g_t(s_u^{\text{text}}), \quad (7)$$

where $g_t(\cdot)$ denotes the transformation layer, which we implement using linear layers. Since $s_u^{\text{text}}$ contains step-by-step reasoning knowledge about user preferences, the model can recommend item with matching item-side information $z_i$ to the user and provide explainable recommendation rationales of natural language form.

## 3 EXPERIMENTS

### 3.1 Experimental Settings

*3.1.1 Datasets.* We conducted our experiments on three categories from the Amazon Review dataset: **Video Games** (Games), **Grocery and Gourmet Food** (Food), and **Home and Kitchen** (Home). Following the pre-processing procedure from [9, 12], and we random sampling 26222 interactions with 3000 users and 9647 items for Games datasets, 17668 interactions with 2000 users and 11190 items for Food datasets, 20000 interactions with 2000 users and 15740 items for Home datasets. More information about these datasets are presented in A.1.

*3.1.2 Baselines.* We adopt three widely used sequential recommendation models as the backbone.

- **GRU4Rec** [7] is a pioneering method that uses Recurrent Neural Networks to model user behavior sequences.
- **SASRec**[12] is typical self-attention based framework designed to capture the user's preferences within a sequence.
- **SRGNN**[30] is graph-based model designed to capture the transition information between items in user sequences.

For each backbone model, we examine the performance of its *Item Feature Extensions*: denoted as **GRU4Rec$^+$**, **SASRec$^+$**, and **SRGNN$^+$**. These extended versions involve concatenating the item ID vector and item description text vector as the input, resulting in enhanced item representations. We also introduce another *ChatGPT Feature Extension* of each backbone model: **SLIM$^-$**. This extended version directly input the rationales generated by the teacher model, ChatGPT, into Eq (5) without distillation. We present the implementation details of each methods in A.2.

*3.1.3 Evaluation Metrics .* We utilize three widely-adopted metrics for evaluation: **NDCG@10**, **Hit Rate@10**, and **Hit Rate@20**. The average scores of 5 runs and the standard deviation are reported. Following the strategy in [12], we randomly sample 100 negative items for each user $u$ and rank these items alongside the ground-truth item. The rankings of these 101 items are then used to evaluate.

### 3.2 Overall Performance

*3.2.1 Improvement over Backbone Models in ID-based scenarios.* Our SLIM is highly flexible and can be integrated with any type of sequential recommendation backbone. Firstly, we evaluate the performance of the SLIM across various backbones. The results of these comparisons are presented in Table 1. We have the following observations: (1) Compared to all backbones and their item feature extensions, the proposed SLIM achieves state-of-the-art (SOTA) performance across all datasets. This further substantiates the effectiveness of our model in enhancing traditional recommendations. Notably, SLIM achieves a relative improvement of 28.17% over the GRU4Rec$^+$ in terms of Hit Rate@10 on the Home dataset. These improvements are attributed to the meaningful rationales generated from our distilled student model, which contains a wealth of knowledge that benefits recommendations as a valuable supplement to closed collaborative signals. (2) Surprisingly, in most cases (22 / 27), SLIM in each backbone outperforms the ChatGPT feature extensions SLIM$^-$, achieving a relative improvement of 12.68% in terms of Hit Rate@10 on the Home dataset with the GRU4Rec backbone.

**Table 1: ID-based scenarios. Comparison of recommendation performance among different backbones. The best results are highlighted in bold. "Improv." indicates the relative improvement of SLIM compared to the best performance in backbones (original backbone and backbone$^{+}$).**

| Methods | Games | | | Food | | | Home | | |
|---|---|---|---|---|---|---|---|---|---|
| | NDCG@10 | Hit @10 | Hit @20 | NDCG@10 | Hit @10 | Hit @20 | NDCG@10 | Hit @10 | Hit @20 |
| GRU4Rec | 17.61 ± 0.18 | 30.87 ± 0.56 | 42.39 ± 0.62 | 9.10 ± 0.30 | 15.27 ± 0.58 | 19.51± 0.24 | 2.19 ± 0.21 | 4.17 ± 0.39 | 7.53 ± 0.64 |
| GRU4Rec$^{+}$ | 27.33 ± 0.53 | 44.06 ± 0.79 | 56.53 ± 1.24 | 17.75± 0.78 | 31.10 ± 1.09 | 45.01 ± 1.46 | 12.19 ± 1.02 | 26.76 ± 2.58 | 49.10 ± 3.82 |
| SLIM$^{-}$ | 27.70 ± 0.47 | 45.13 ± 0.56 | 57.70 ± 0.37 | 17.97 ± 0.70 | 31.78 ± 1.47 | 46.88 ± 2.10 | 13.59 ± 1.05 | 30.30 ± 2.01 | 55.85 ± 3.55 |
| SLIM | **28.37 ± 0.41** | **45.68 ± 0.53** | **58.09 ± 0.58** | **18.32 ± 0.53** | **32.56 ± 1.30** | **46.92 ± 1.82** | **15.64 ± 0.51** | **34.33 ± 1.53** | **62.93 ± 3.46** |
| Improv. | 3.81% | 3.68% | 2.76% | 3.21% | 4.69% | 4.24% | 28.3% | 28.29% | 28.17% |
| SASRec | 22.73 ± 0.28 | 37.77 ± 0.52 | 51.53 ± 0.39 | 26.78 ± 0.24 | 35.78 ± 0.36 | 43.32 ± 0.55 | 2.66 ± 0.22 | 5.56 ± 0.72 | 14.93 ± 1.53 |
| SASRec$^{+}$ | 27.46 ± 0.19 | 44.88 ± 0.63 | 58.90 ± 0.38 | 30.95 ± 0.38 | 44.98 ± 0.53 | 55.61 ± 1.12 | 5.58 ± 0.10 | 11.09 ± 0.16 | 20.69 ± 0.77 |
| SLIM$^{-}$ | **31.58 ± 0.35** | 50.83 ± 0.62 | 63.45 ± 0.71 | 32.65 ± 0.15 | 48.01 ± 0.48 | 59.25 ± 0.56 | 5.95 ± 0.32 | 11.83 ± 0.62 | **22.43 ± 0.54** |
| SLIM | 31.43 ± 0.39 | **51.11 ± 0.82** | **64.10 ± 0.26** | **32.80 ± 0.40** | **48.27 ± 0.64** | **59.30 ± 0.89** | **6.01 ± 0.19** | **12.01 ± 0.38** | 22.29 ± 0.85 |
| Improv. | 14.46% | 13.88% | 8.83% | 5.98% | 7.31% | 6.64% | 7.71% | 8.3% | 7.73% |
| SRGNN | 16.45 ± 0.22 | 29.29 ± 0.14 | 40.99 ± 0.52 | 10.99 ± 2.07 | 20.32 ± 4.30 | 32.14 ± 6.55 | 5.04 ± 0.83 | 13.48 ± 2.23 | 37.22 ± 3.85 |
| SRGNN$^{+}$ | 21.54 ± 0.64 | 36.77 ± 1.05 | 49.11 ± 1.54 | 11.91 ± 0.71 | 21.39 ± 1.91 | 33.63 ± 3.41 | 11.61 ± 1.14 | 25.22 ± 2.46 | 43.85 ± 3.58 |
| SLIM$^{-}$ | 22.35 ± 1.48 | 37.69 ± 1.53 | 51.29 ± 0.59 | **12.92 ± 0.78** | **23.80 ± 1.60** | **37.22 ± 2.75** | 11.25 ± 1.42 | 24.28 ± 3.19 | 44.05 ± 5.97 |
| SLIM | **23.77 ± 0.20** | **39.81 ± 0.52** | **52.34 ± 0.63** | 12.38 ± 0.51 | 22.98 ± 1.30 | 36.44 ± 1.72 | **12.29 ± 1.39** | **26.51 ± 2.71** | **47.01 ± 3.61** |
| Improv. | 10.35% | 8.27% | 6.58% | 3.95% | 7.43% | 8.36% | 5.86% | 5.11% | 7.21% |

**Table 2: ID-agnostic Text Matching model. Comparison of recommendation performance without relying on any backbone models. SLIM-Step$i$ indicates only using the $i\text{-}th$ step rationales generated by SLIM. SLIM$^{-}$ is the ChatGPT feature extension of this model.**

| Methods | Games | | | Food | | | Home | | |
|---|---|---|---|---|---|---|---|---|---|
| | NDCG@10 | Hit @10 | Hit @20 | NDCG@10 | Hit @10 | Hit @20 | NDCG@10 | Hit @10 | Hit @20 |
| SLIM-Step1 | 13.78 ± 0.59 | 26.08 ± 0.91 | 41.71 ± 0.62 | 13.62 ± 0.22 | 24.90 ± 0.59 | 38.15 ± 0.69 | 4.25 ± 0.06 | 9.53 ± 0.27 | 18.91 ± 0.77 |
| SLIM-Step2 | 16.78 ± 0.66 | 30.09 ± 0.89 | 45.75 ± 0.59 | 13.71 ± 0.48 | 24.23 ± 0.78 | 36.89 ± 0.77 | 4.75 ± 0.29 | 10.30 ± 0.55 | 19.99 ± 0.61 |
| SLIM-Step3 | 20.20 ± 0.31 | 35.57 ± 0.54 | 50.04 ± 0.47 | 15.69 ± 0.26 | 26.69 ± 0.68 | 39.19 ± 0.98 | **4.83 ± 0.27** | **10.39 ± 0.38** | **21.05 ± 0.43** |
| SLIM$^{-}$ | 21.75 ± 0.58 | 38.05 ± 0.88 | **53.73 ± 0.79** | **19.08 ± 0.71** | **32.34 ± 0.74** | **44.63 ± 0.90** | 4.63 ± 0.28 | 10.15 ± 0.62 | 20.11 ± 1.01 |
| SLIM | **21.99 ± 0.22** | **38.33 ± 0.32** | 53.59 ± 0.72 | 18.13 ± 0.5 | 30.84 ± 0.54 | 44.04 ± 0.66 | 4.49 ± 0.24 | 9.96 ± 0.47 | 19.67 ± 0.59 |

While SLIM's knowledge is distilled from the teacher model Chat-GPT, the lack of control over closed-source models may result in the generation of correct but irrelevant responses to recommendations. This indicates that our smaller model can further prioritize the information relevant to recommendations after distillation. Despite being smaller in scale, it greater effectiveness in recommendations.

3.2.2 *Performance in ID-agnostic scenarios.* To establish a more efficient and generalizable model, we evaluate the performance of SLIM in ID-agnostic scenarios, i.e., we solely based on matching CoT-based sequence embeddings and text-based item embeddings as Eq (7), named Text Matching. The results are shown in Table 2. We also obtained interesting findings: (1) In comparison to models that generate rationales based on the teacher model (SLIM$^{-}$), SLIM outperforms it in 55.56% of cases, despite having only 4% of the parameters compared to ChatGPT. This demonstrates that even with limit training samples (1000-2000) and a smaller model size, SLIM can generate high-quality recommendation rationales that are highly competitive with ChatGPT. (2) Additionally, this straightforward matching approach exhibits superior performance compared to all ID-based backbones listed in Table 1. This indicates that high-quality text from both the sequence and item side can lead to promising recommendations, even without meticulous design of

the text encoder. (3) To verify the effectiveness of each step in the rationales, i.e., the user interest of Step1, the item category of Step2, and the specific product of Step3, we evaluate them separately in Text Matching. It is worth noting that the ranking of recommendation performance consistently follows the pattern of Step3 > Step2 > Step1 in all cases. Surprisingly, on the Home dataset, Step3 even surpasses the performance achieved using the entire Rationale. These results suggest that the smaller model trained with CoT prompting is capable of step-by-step thinking, similar to human reasoning. As the chain of thought evolves, the information relevant to recommendations will be inferred. However, the performance of the Step1 is not satisfactory, possibly because the macroscopic information in this step fails to align well with the microscopic information on the item side, such as titles and categories. Nevertheless, the first step still plays a crucial role as the foundation for subsequent reasoning processes and ensures the interpretability of the model.

## 3.3 Merits of SLIM

**Potentially Good Interpretability for the Recommendation Results.** Figure 4 illustrates a sample where SLIM successfully recommends the ground-truth, while SASRec fails. The target next item in this sample is a long-tail item that only appears once in the

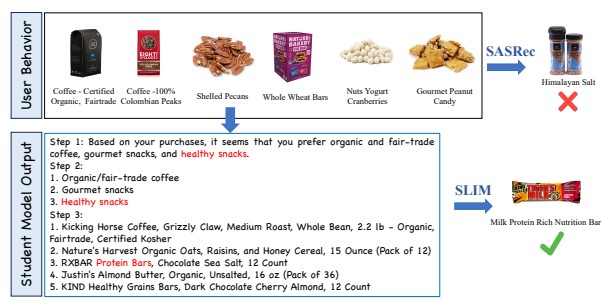

Figure 4: Case study. Comparison of the predictions for the next item obtained from SASRec and SLIM.

training set. As a result, traditional ID-based models struggle to capture adequate collaborative signals. However, SLIM's generated rationales are able to deduce the user's preferences, which align closely with the characteristics of the target item "Milk Protein Rich Nutrition Bar", such as the categories of "Healthy snacks". More significantly, SLIM showcases its remarkable reasoning capabilities and extensive domain knowledge by accurately inferring that users are likely to purchase "Protein Bars". In this manner, the textual information from both the sequence side and item side aligns well, leading to a high similarity in the vector space. Moreover, SLIM generates rationales in human-understandable natural language. The rationales provided in Step 1 and Step 2 offer justifications for the recommendation of "Protein Bars" by SLIM. For each recommended item, SLIM can provide a natural language explanation, enhancing the interpretability of the recommendation process.

**Consistent Improvement for User with Different Sparsity.** To investigate the impact of interaction data sparsity, we group users based on the sparsity level of their interactions and evaluate the performance of SLIM separately on different user groups. Specifically, we sorted users based on their interaction frequency, and then divided them equally into five user groups. Subsequently, SLIM and SASRec are trained separately on the interaction data of each user group, and compare their recommendation performance on different user groups. The results, as depicted in Figure 5, that SLIM consistently outperforms SASRec across all user groups, and SLIM exhibits greater improvement on the relatively sparse user group $G1$ compared to the dense user group $G5$. This suggests that our method's improvement is stable and robust, effectively mitigating the issue of sparsity in sequential recommendation.

**Impressive Capability of Alleviating Popularity Bias.** In the field of recommender systems, popularity bias means that popular items are recommended even more frequently than their popularity would warrant. This bias intensifies the long-tail effects in real-world recommendation domains. To analyze the impact of our proposed SLIM on popularity bias, we count the frequency of items in the training data and recommendation results. As shown in Figure 6, compared with the traditional method SASRec to recommend the head items with high popularity, our method can effectively recommend the tail items. Experimental results confirm that our method significantly mitigate the popularity bias.

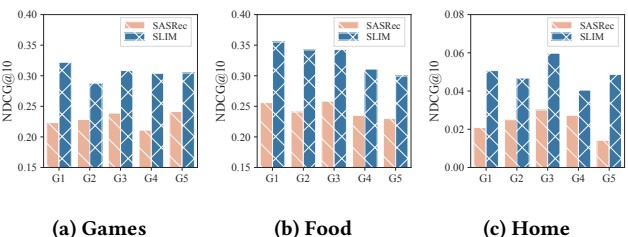

(a) Games      (b) Food      (c) Home

Figure 5: Analysis for different sparse-level users. The sparsity decreases from user group $G1$ to user group $G5$.

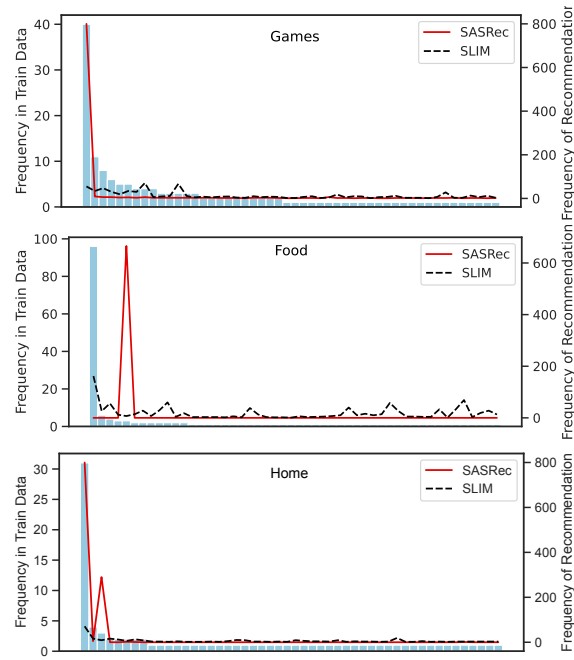

Figure 6: Analysis of popularity bias. We sort the items based on their frequency in the training set (*i.e.,* popularity) and draw line plots based on each item's frequency in the recommendation results of SASRec and SLIM, respectively.

**Significantly More Affordable Compared to SOTAs.** SLIM has demonstrated promising performance. In this part, we will analyze the efficiency of the model through a comprehensive cost analysis across multiple dimensions, including time cost, model size, deployment difficulty and API monetary cost. Specifically, we compare two representative recommendation models based on the generation capabilities of LLM. (1) LLM as a ranker [9]. (2) LLM as a knowledge enhancer [31]. We do not take into account the costs associated with backbones, as the backbone model is typically variable and the cost is generally negligible compared to LLM. Due to the two-stage nature of the LLM as a knowledge enhancer approach, it involves offline knowledge generation based on LLM and online inference. The term "Offline/Online Time" refers to the average response time of the closed-source ChatGPT API for the compared methods. Conversely, our method corresponds to the average inference time on

**Table 3: The comparison of LLM costs. "Offline Time" represents the time it takes for LLM to generate one piece of knowledge offline. "Online Time" represents the time it takes for LLM to perform inference for each ranking online.**

| Costs | LLM as Ranker | LLM as Enhancer | SLIM |
|---|---|---|---|
| Offline Time (s) | ✗ | 5.54 (API call) | 6.11 |
| Online Time (s) | 5.54 (API call) | ✗ | ✗ |
| Model Size | 175B | 175B | 7B |
| Deployment | Hard | Hard | 1 A100 |
| API Costs/Input | $0.0015/1K tokens | $0.0015/1K tokens | ✗ |
| API Costs/Output | $0.002/1K tokens | $0.002/1K tokens | ✗ |

a single Nvidia A100 GPU. It is worth mentioning that despite deploying SLIM on only one GPU, we achieve a comparable time cost compared to the API call duration of ChatGPT, which requires significant resource consumption for deployment. From Table 3, it can be concluded that SLIM is a highly efficient model compared to existing LLM-based recommendations. It possesses acceptable inference latency, minimal model size, and can be deployed on limited resources. Additionally, being based on an open-source model, it does not incur any financial cost.

On the other hands, we also study the data efficiency of fine-tuning the student models, and we conclude that only 1000 samples is enough for a promising performance. We present the details of the analysis in B.1.

## 4 RELATED WORK

### 4.1 Sequential Recommender Systems

The core idea of existing sequential recommendation models lies in initially formalizing user behavior as a chronologically-ordered interaction sequence with items, followed by designing diverse behavior encoders to learn behavior patterns that accurately depict user interests [26]. GRU4Rec [7] is one of the earliest attempts to learn evolving patterns for user behaviors using Gated Recurrent Units (GRU). With the rapid development of deep learning, there have also been emerging many neural network architectures as behavior encoders, including Convolutional Neural Networks-based methods [23], Attention-based methods [12], and Graph Neural Networks-based methods [30]. To enhance the transferability of the sequence modeling, recent studies have begun exploring ID-agnostic text-based modeling approaches [8, 13], such as Recformer [13], which proposes formulating each item as a "sentence" and designing a Transformer-based language model to learn user preference representations. However, these methods rely on limited textual information provided by the recommendation dataset, which restricts the model's capabilities due to its isolation from rich open-world knowledge. Recently, the emergence of LLM that utilize massive training corpora and large model sizes has disrupted the traditional closed-loop of user-item interaction in recommendations [15, 16].

### 4.2 LLM Enhanced Recommender Systems

The utilization of LLMs, with their human-like understanding and generation capabilities, introduces new knowledge spaces in recommendations [5, 14, 29]. To integrate LLM's generation capabilities into recommendations, the current methods can be primarily categorized into two mainstream trends based on the different roles LLM plays in the recommendation pipeline. The first trend involves utilizing LLM as a ranker or scorer [2, 9, 15, 32, 33]. For instance, [9] explores the zero-shot ranking capabilities of LLM in recommendation. This requires careful design of prompts that involve a predefined list of candidate items for the limited re-ranking stage. [33] proposes to view recommendation as instruction following by LLMs. In this approach, 39 instruction templates are manually designed to enable LLMs to execute the instructions. However, these methods often exhibit limited performance because the frozen LLMs are typically trained on open-world corpora that lack domain-specific collaborative signals from recommendations. To incorporate collaborative information, recent studies have started exploring another trend, which involves utilizing LLM as a knowledge enhancer to complement traditional recommender systems [16, 31]. For example, [31] explores the acquisition of user preferences and item factual knowledge from ChatGPT, and utilizes them to enhance traditional Click-Through Rate (CTR) prediction. [16] proposes to employ open-source LLM as content encoders and utilize closed-source ChatGPT to enrich the training data from various perspectives. While promising, existing work has not fully leveraged the step-by-step reasoning capabilities of LLM in the recommendation scenario. Furthermore, current approaches often rely on the use of large model sizes to achieve improved reasoning capabilities. Although techniques like prestoring can be employed to deploy only the inference model, these models still require larger model sizes in either offline or online stages, which may not be feasible in real-world recommender systems.

## 5 CONCLUSION

In this paper, we propose SLIM, a method that enables sequential recommender systems to leverage the substantial reasoning capabilities of LLMs in a resource-efficient manner. We design a step-by-step knowledge distillation module to transfer the step-by-step reasoning capabilities in recommendation from a larger teacher model to a smaller student model (with approximately 4% of the parameters of the teacher model). This smaller model evolves into a proficient reasoner, which can be directly deployed as a "slim" knowledge generator for sequential recommendation. Consequently, this knowledge can be flexibly integrated with any sequential recommendation backbone and utilized in both ID-based and ID-agnostic scenarios. The experimental results demonstrate that SLIM significantly improves the performance of sequential recommendation backbones. It also achieves promising results in ID-agnostic scenarios without relying on any backbone. Furthermore, additional analysis experiments highlight that the costs associated with SLIM are affordable and have the potential to enhance the interpretability of recommendations. A possible future direction is to design customized knowledge encoders to further capture the information from smaller models.

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

# A DETAILS OF EXPERIMENTAL SETTINGS

## A.1 Dataset

We conducted our experiments on three categories from the Amazon Review dataset: **Video Games** (Games), **Grocery and Gourmet Food** (Food), and **Home and Kitchen** (Home). In this dataset, reviews are considered as interactions. We follow the same preprocessing procedure from [9, 12], i.e., users and items with fewer than five interactions were filtered out. The interactions of each user were sorted in ascending order based on timestamps to construct the corresponding historical interaction sequences. To cope with the significant cost associated with ChatGPT, we conducted random sampling of complete user interactions from each dataset. Following the sampling process, the resulting sizes of the Games, Food, and Home datasets are as follows: the number of users [3000, 2000, 2000], the number of items [9647, 11190, 15740], and the number of interactions [26222, 17668, 20000]. For item description text, we utilized the titles from all datasets and the sales type from the Food and Home datasets. Following [9, 12], we employ the leave-one-out strategy. Specifically, for each interacted sequence $S^u$, the most recent interaction is used for testing, the second most recent interaction for validation, and all remaining interactions for training. During testing, the input sequences consist of both training and validation interactions.

## A.2 Implementation Details

We employ the powerful close-source ChatGPT as the teacher model and the open-source LLaMA2-7B [24] as the student model. The OpenAI's API gpt-3.5-turbo[1] is utilized for generating the rationales of the teacher model. We follow the parameters provided in llama-recipes[2] for fine-tuning LLaMA2-7B. Specifically, for the Games, Food, and Home datasets, data sizes of 1000, 2000, and 2000 were used for fine-tuning LLaMA2-7B, respectively. Additionally, we set the maximum length of generated tokens to be 300 during the inference process for LLaMA2-7B. All embeddings obtained from Eq. (5) have a dimensionality of 768. The implementation of SASRec is based on the PyTorch version code[3]. For GRU4Rec and SRGNN, we utilize the RecBole [4], which is a widely-used open-source recommendation library. The implementation of SLIM ensures that the embedding dimensions remain consistent with the original backbone. We carefully search hyper-parameters of all the baselines to get the best performance.

# B MORE EXPERIMENTAL RESULTS

## B.1 Impact of Different Data Size for Fine-tuning Student Models

In this part, we will analyze the efficiency of the model by analyzing the data size for fine-tuning the student models. In particular, we validate the optimal fine-tuning data size required for student models under different backbones, as illustrated in Figure 7. The recommendation performance demonstrates a trend of initially increasing and then decreasing with an increase in the data size across

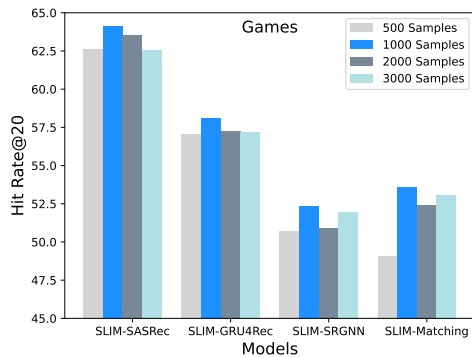

**Figure 7: Compare the recommendation performance of fine-tuning student models with different data sizes using various backbones on Games dataset.**

all backbones. Remarkably, the optimal performance is consistently achieved with 1000 fine-tuning samples. This finding suggests that SLIM can effectively perform the fine-tuning process using minimal data from the teacher model. This is also supported by the observation that fine-tuning based on CoT is more efficient compared to fine-tuning based on labels [17].

---

