# OpenReview forum: "Can Small Language Models be Good Reasoners in Recommender Systems?"
_ACM.org/TheWebConf/2024/Conference — TheWebConf24 Oral_

### Official Review · Reviewer_GAPb · 2023-11-21

**Novelty:** 6
**Technical Quality:** 6

**Review:**

This paper is well formulated and addresses the problem exactly what I previously concerned with. That is, recent works, adopt LLMs as the knowledge enhancer to augument the traditional RS mode, should heavily rely on the open-world knowledge and emergent abilities (e.g., reasoning) possessed by LLMs. Therefore, they usually utilize gpt-3.5-turbo as the knowledge enhancer, which can lead to unacceptable cost when it comes with industrial-scale users & items.

Therefore, in this paper, authors propose to distill and align a small LLM (i.e., LLaMA 7B) according to the output of a big LLM (i.e., gpt-3.5-turbo). In this way, we are able to obtain a small LLM that is capable of playing the role as knowledge enhancer for the downstream RS models.

In summary, this is a nice and sound paper for me, but I also got some additional concerns for authors, which would be listed in the Questions section below.

**Questions:**

I got the following major concerns for this paper:

1. In Table 1 & 2, it is surprising or even werid that gpt-3.5-turbo version (i.e., SLIM-) fails to defeat small LLM version (i.e., SLIM). Although authors explain that some irrelevant content would be generated by the teacher model, I expect further analysis and case study on this phenomenon and conjecture.
2. The generalization ability of the knowledge distillation small LLM should be carefully analyzed. That is, once we align small LLM to the output of big LLM, can we generalize this knowledge enhancer to other recommedation domains (e.g., other datasets)? If not, we have to distill for each dataset, which means that we have to call gpt-3.5-turbo every time when a new dataset comes. This actually does not save the cost, if the knowledge enhancer got no generalization abilities. I would expect experiments like this: distill and align a small LLM on dataset A, and then directly apply it on dataset B to see whether A-based knowledge enhancer can promote the performance in B.
3. I agree with Reviewer JJKQ that the title "small language model" could be confusing and misleading since LLaMA-7B is not small. I suggest maybe **"Small" Large Language Model"** is more suitable.

I would like to increase my scores if the two key concerns above is well tackled.

Also, another minor suggestion is provided here. In line 85-89, authors claim that zero-shot or few-shot LLM as a ranker is hard to defeat traditional RS models. However, recent works [1, 2] actually show that with designed training strategy, pure few-shot LLMs are able to outperform traditional RS models. Better discuss with these related works.

[1] ReLLa: Retrieval-enhanced Large Language Models for Lifelong Sequential Behavior Comprehension in Recommendation

[2] Do LLMs Understand User Preferences? Evaluating LLMs On User Rating Prediction

**Reviewer Confidence:**

4: The reviewer is certain that the evaluation is correct and very familiar with the relevant literature

**Scope:**

4: The work is relevant to the Web and to the track, and is of broad interest to the community

---

### Official Review · Reviewer_JJKQ · 2023-11-21

**Novelty:** 4
**Technical Quality:** 5

**Review:**

This paper proposes a distillation-based method to distill information from a black box very large LM into a relatively small LLM for recommendations.

This paper has the following merits that I appreciate very much.

+ Distillation is a good idea to reduce the cost of inference when applying LLM in real-world recommendations.
+ The proposed CoT prompting and teacher-student model training strategy makes sense.
+ Experiments, to some extent, demonstrate the effectiveness of the proposed method.

My main concerns are as follows.

- First, the "small language model" in the title seems misleading. Llama-7B is absolutely not a small language model, even though it is smaller than the super huge GPT-4 model.
- The distillation process seems to follow the traditional model distillation process with student-teacher architecture, which seems to lack technical insights.

**Questions:**

Please refer to my concerns in the main review.

**Reviewer Confidence:**

3: The reviewer is confident but not certain that the evaluation is correct

**Scope:**

4: The work is relevant to the Web and to the track, and is of broad interest to the community

---

### Official Review · Reviewer_Ehq3 · 2023-11-28

**Novelty:** 5
**Technical Quality:** 5

**Review:**

The paper introduces a Step-by-step knowledge distillation framework for recommendation (SLIM), designed to enable sequential recommenders to leverage the reasoning capabilities of Large Language Models (LLMs) in a resource-efficient manner. The framework incorporates CoT prompting based on user behavior sequences for a larger teacher model. Rationales generated by the teacher model serve as labels to distill a smaller student model (e.g., LLaMA2-7B), allowing the student model to acquire step-by-step reasoning capabilities in recommendation tasks. The generated rationales from the student model are encoded into a dense vector, enhancing recommendation performance in both ID-based and ID-agnostic scenarios.

Pros:

1. It is interesting to apply LLM for recommendation in an efficient and effective way.

2. The experimental results show good performance improvement.

3. This paper is well written and easy to understand.

Cons:

1. Only Amazon Review dataset is used for experiment, which is not enough.

2. The baseline methods compared in this paper is weak and insufficient. More baseline methods especially those also incorporated LLMs should be compared.

3. It is not clear whether the proposed method can generalize to different LLMs.

**Questions:**

Why not try more datasets and LLMs?

Why not compare with stronger baselines?

**Reviewer Confidence:**

2: The reviewer is willing to defend the evaluation, but it is likely that the reviewer did not understand parts of the paper

**Scope:**

3: The work is somewhat relevant to the Web and to the track, and is of narrow interest to a sub-community

---

### Official Review · Reviewer_85tv · 2023-12-11

**Novelty:** 6
**Technical Quality:** 6

**Review:**

The authors propose a Step-by-step knowLedge dIstillation fraMework (SLIM) that uses smaller models (e.g., LLaMA2-7B) to distill knowledge from larger "teacher" models. This process involves generating rationales through chain-of-thought (CoT) prompting based on user behavior, which are then used to train the smaller "student" models. The distilled student models hence acquire the step-by-step reasoning for recommendations. Rationales can further be encoded into dense vectors useful for both ID-based and ID-agnostic recommendation scenarios.
Strengths:
SLIM retains the strengths of LLMs without requiring extensive resources, making it more practical for real-world applications.

SLIM's effectiveness and cost-efficiency is demonstrated over state-of-the-art baselines through extensive experiments​​.

The proposed approach can work in both ID-based and ID-agnostic scenarios.
The experiments explore various facets of recommendation not limited to accuracy, this includes the effect of ID based /ID agnostic, data sparsity and popularity bias, as well as the efficiency and the LLM cost in offline and online times.

Mild Weakness that was later  addressed by author response:
Popularity bias is evaluated however no popularity bias metrics are used; also, the methods being compared do not address this bias, why not compare to a baseline that takes this bias into account such as “Debiasing the Cloze Task in Sequential Recommendation with Bidirectional Transformers”, in KDD 2022 by Damak et al.

**Questions:**

the plots for popularity bias are informative, however why not also compute standard metrics for this bias?

**Reviewer Confidence:**

4: The reviewer is certain that the evaluation is correct and very familiar with the relevant literature

**Scope:**

4: The work is relevant to the Web and to the track, and is of broad interest to the community

---

### Decision · Program_Chairs · 2024-01-22

**Decision:**

Accept (Oral)

**Comment:**

A step-by-step knowledge distillation framework based on LLMs is proposed in this paper to improve the performance of sequential recommendations. All reviewers agreed that the paper is well written and the proposed method is interesting and sound. For the weakness, they mainly raised concerns about the paper's title, the datasets, the baselines, the model's generalizability, and the analysis of the results. After the rebuttal phase, some reviewers changed their opinions and raised the scores. If the paper is accepted, it is expected that the authors' responses will be reflected in their revision.